# Enhanced spin Seebeck effect via oxygen manipulation

Jeong-Mok Kim [1,6], Seok-Jong Kim [1,2,6], Min-Gu Kang[1,5,6], Jong-Guk Choi[1], Soogil Lee[1], Jaehyeon Park[2], Cao Van Phuoc [3], Kyoung-Whan Kim [4], Kab-Jin Kim[2], Jong-Ryul Jeong [3], Kyung-Jin Lee [2] ✉ & Byong-Guk Park [1] ✉

Spin Seebeck effect (SSE) refers to the generation of an electric voltage transverse to a temperature gradient via a magnon current. SSE offers the potential for efficient thermoelectric devices because the transverse geometry of SSE enables to utilize waste heat from a large-area source by greatly simplifying the device structure. However, SSE suffers from a low thermoelectric conversion efficiency that must be improved for widespread application. Here we show that the SSE substantially enhances by oxidizing a ferromagnet in normal metal/ferromagnet/oxide structures. In W/CoFeB/AlO$_x$ structures, voltage-induced interfacial oxidation of CoFeB modifies the SSE, resulting in the enhancement of thermoelectric signal by an order of magnitude. We describe a mechanism for the enhancement that results from a reduced exchange interaction of the oxidized region of ferromagnet, which in turn increases a temperature difference between magnons in the ferromagnet and electrons in the normal metal and/or a gradient of magnon chemical potential in the ferromagnet. Our result will invigorate research for thermoelectric conversion by suggesting a promising way of improving the SSE efficiency.

The thermoelectric effect that converts heat to electricity in solid-state devices is attracting much attention as a promising candidate for a carbon-free power generation from waste heat[1]. Thermoelectric generation based on conventional Seebeck effect employs a longitudinal geometry[2], in which an electric voltage is generated along a temperature gradient $\nabla T$ (Fig. 1a). The longitudinal geometry is not favored for applications because it requires a thermopile composed of multiple and alternatively connected thermoelectric materials of different types[3], which is complex to cover a large-area heat source. This limitation can be overcome by a thermoelectric device in transverse geometry using the spin Seebeck effect (SSE)[4–8] that generates an electric voltage in the direction perpendicular to a temperature gradient[9] (Fig. 1b). For practical applications, however, a thermoelectric conversion efficiency of SSE is insufficient and must be substantially enhanced.

In normal metal (NM)/ferromagnet (FM) heterostructures (Fig. 1b), a basic building block for SSE, the thermoelectric conversion via longitudinal SSE consists of three separate processes; First, a vertical temperature gradient generates a temperature difference $T_{m-e}$ between magnons in FM and electrons in NM (described by the magnon temperature model[10–12]) and a magnon spin current associated with a magnon chemical potential gradient $\nabla \mu_m$ in FM (described by the magnon drift-diffusion model[13,14]). Second, a thermal spin pumping from FM to NM occurs. Third, a spin current generates a transverse voltage via inverse spin Hall effect (ISHE) of NM. These three processes can be independently controlled so that the enhancement of SSE in each process is multiplied. Previous studies have focused mainly on the improvement of the second process by increasing the spin-mixing conductance at the NM/FM interface[15–17] and the third one by employing NM with large spin Hall

[1]Department of Materials Science and Engineering, KAIST, Daejeon 34141, Korea. [2]Department of Physics, KAIST, Daejeon 34141, Korea. [3]Department of Materials Science and Engineering, Chungnam National University, Daejeon 34134, Korea. [4]Center for Spintronics, Korea Institute of Science and Technology, Seoul 02792, Korea. [5]Present address: Department of Materials, ETH Zurich, Zurich, Switzerland. [6]These authors contributed equally: Jeong-Mok Kim, Seok-Jong Kim, Min-Gu Kang. ✉e-mail: kjlee@kaist.ac.kr; bgpark@kaist.ac.kr

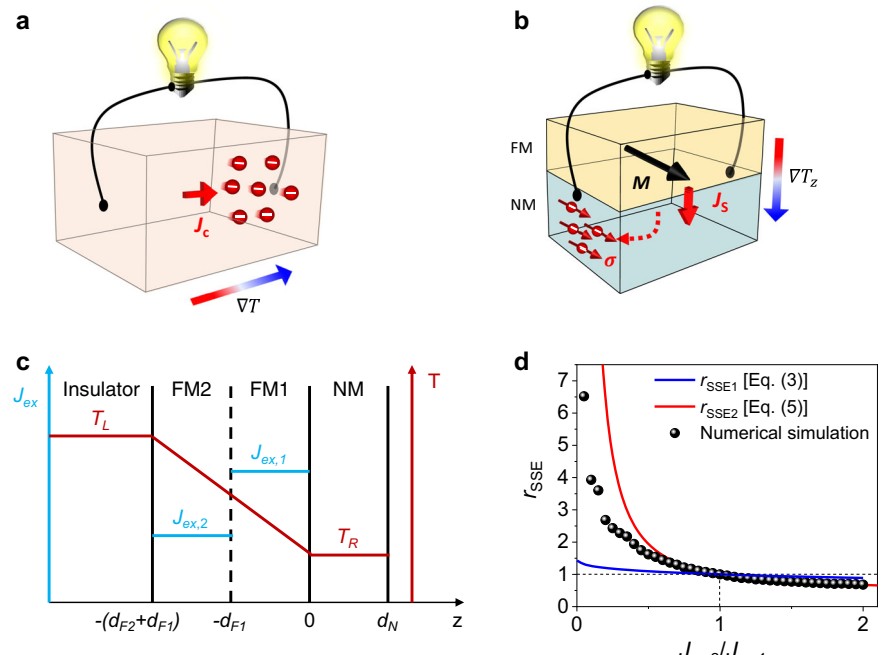

**Fig. 1 | Schematics of SSE and theoretical results. a** Process of conventional Seebeck effect where $\nabla T$ and $J_c$ denote the temperature gradient and charge current, respectively. **b** Process of SSE in FM/NM structure where $\nabla T_z$, $M$, $J_s$, and $\sigma$ denote the vertical temperature gradient, magnetization, spin current, and spin polarization vector, respectively. **c** Model structure where FM consists of two ferromagnetic layers with different exchange interaction $J_{ex}$. **d** Theoretical and numerical results of the ratio $r_{SSE}$ (defined in the main text) as a function of $J_{ex,2}/J_{ex,1}$.

angle[18,19]. In addition, lowering the damping of FM[20] for the first process or reducing the thermal conductivity of FM[21] enhances the SSE efficiency by several factors.

In this work, we report an efficient way to enhance the SSE signal through the first process by reducing the exchange interaction $J_{ex}$ at the hotter region of FM. As described below, a reduction of $J_{ex}$ increases a magnon heat current, which in turn increases $T_{m-e}$ and thus the SSE signal. Moreover, as $\nabla \mu_m$ contributes to a magnon spin current, the SSE can enhance by increasing $\nabla \mu_m$ at a given temperature gradient. Following the Bloch's $T^{3/2}$ law, the most important factor for the magnon chemical potential is $J_{ex}$ because the magnon density at a given temperature is determined by the Curie temperature. Therefore, a reduction of $J_{ex}$ at the hotter region of FM increases a magnon spin current, which also enhances the SSE signal.

In what follows, we discuss two different theoretical mechanisms to describe the SSE based on the magnon temperature model[10–12] and the magnon drift-diffusion model[13,14]. Both mechanisms may coexist[14] but we consider these separately for simplicity. We first describe the magnon temperature model[10–12] for an insulator/FM1/FM2/NM structure where $J_{ex}$ differs between FM1 and FM2 (Fig. 1c). From the thermal circuit model equivalent to the layered structure of Fig. 1c and ignoring magnon relaxation, the SSE voltage $V_{SSE}$ is found to be proportional to (Supplementary Note 1):

$$V_{SSE} \propto T_{m-e} = \frac{Q_1^{FM} R_1^{FM} + Q_2^{FM} R_2^{FM}}{R_1^{FM} + R_2^{FM} + R_{F2|F1}^{int} + R_{F1|N}^{int}} R_{F1|N}^{int} \qquad (1)$$

where $Q_i^{FM}$ is the magnon heat current of FM$i$ ($i = 1, 2$), $R_i^{FM}$ is the magnon heat resistance of FM$i$, and $R_{F2|F1}^{int}$ ($R_{F1|N}^{int}$) is the interface magnon heat resistance at the FM2/FM1 (FM1/NM) interface. Assuming no loss of magnon heat current at the FM2|FM1 interface (i.e.,

$R_{F2|F1}^{int} \to 0$) and $R_1^{FM} + R_2^{FM} \gg R_{F1|N}^{int}$, Eq. (1) is simplified as

$$V_{SSE} \propto \frac{Q_1^{FM} R_1^{FM} + Q_2^{FM} R_2^{FM}}{R_1^{FM} + R_2^{FM}} R_{F1|N}^{int}. \qquad (2)$$

Using $Q_i^{FM} = \kappa_i \nabla T$ and $R_i^{FM} = d_{Fi}/(\kappa_i A)$ with the magnonic heat conductivity $\kappa$, the temperature gradient $\nabla T$, the FM thickness $d_F$, and the area $A$ of the structure, $V_{SSE}$ is found to be proportional to $\kappa_1 \kappa_2 (d_{F1} + d_{F2})/(\kappa_1 d_{F2} + \kappa_2 d_{F1})$. Given $\kappa_i \propto (J_{ex,i})^{-1/2}$ [14] (Supplementary Table 1), the ratio $r_{SSE1}$ of $V_{SSE}$ with $J_{ex,1} \neq J_{ex,2}$ to $V_{SSE}$ with $J_{ex,1} = J_{ex,2}$ for the magnon temperature model is given as

$$r_{SSE1} = \frac{d_{F1} + d_{F2}}{d_{F1} + d_{F2}\sqrt{\frac{J_{ex,1}}{J_{ex,2}}}}. \qquad (3)$$

Equation (3) shows that $r_{SSE1}$ is larger than 1 for $J_{ex,2} < J_{ex,1}$ (Fig. 1d). It is because a reduced $J_{ex}$ at the hotter region of FM (i.e., FM2) decreases $R_2^{FM}$, which in turn increases $V_{SSE}$.

We next describe the magnon drift-diffusion model[13,14] for the same structure of Fig. 1c. Ignoring $T_{m-e}$ and solving magnon diffusion equations in FMs coupled with spin diffusion equation in NM gives $V_{SSE}$ as (Supplementary Note 1).

$$V_{SSE} = \theta_{SH} \nabla T \frac{2 e l_N d \left[ \left( d_{F1}^2 l_{F2}^2 L_1 + d_{F2}^2 l_{F1}^2 L_2 \right) \sigma_{F1} + 2 d_{F1} d_{F2} l_{F1}^2 L_1 \sigma_{F2} \right]}{\hbar d_N l_{F1}^2 l_{F2}^2 \sigma \sigma_{F1}} \operatorname{csch} \frac{d_N}{l_N} \sinh^2 \frac{d_N}{2 l_N}, \qquad (4)$$

where $\theta_{SH}$ is the spin Hall angle of NM, $l_N$ is the spin diffusion length of NM, $l_{Fi}$ is the magnon diffusion length of FM$i$, $\sigma$ is the charge conductivity of NM, $\sigma_{Fi}$ is the magnon spin conductivity of FM$i$, $L_i$ is the spin Seebeck coefficient of FM$i$, $d_N$ is the NM thickness, and $d$ is the

electrode distance to measure $V_{SSE}$. We obtain Eq. (4) with assumptions of continuous magnon chemical potential at the FM2/FM1 interface and $d_{F1(2)} \ll l_{F1(2)}$ to simplify $V_{SSE}$ (Supplementary Note 1). Given $\sigma_F \propto L \propto J_{ex}^{-1/2}$ and $l_F \propto J_{ex}^{1/2}$ [14] (Supplementary Table 1), we obtain the ratio $r_{SSE2}$ of $V_{SSE}$ with $J_{ex,1} \neq J_{ex,2}$ to $V_{SSE}$ with $J_{ex,1} = J_{ex,2}$ for the magnon drift-diffusion model as

$$r_{SSE2} = \left[1 + \left(\left(\frac{d_{F2}}{d_{F1}}\right)^2 + 2\frac{d_{F2}}{d_{F1}}\right)\left(\frac{J_{ex,1}}{J_{ex,2}}\right)^{\frac{3}{2}}\right]\left(1 + \frac{d_{F2}}{d_{F1}}\right)^{-2}. \quad (5)$$

Similar to $r_{SSE1}$ [Eq. (3)], $r_{SSE2}$ is larger than 1 for $J_{ex2} < J_{ex1}$ (Fig. 1d). It is because a reduced $J_{ex}$ at the hotter end of FM increases a gradient of magnon chemical potential $\nabla \mu_m$, which in turn increases $V_{SSE}$.

Since the above analytic theories are obtained with several crude approximations, we also carry out numerical simulations based on the stochastic Landau-Lifshitz-Gilbert equation for thermal spin pumping (Supplementary Note 2), following the procedure of Ref. 22. Numerical results (symbols in Fig. 1d) show a qualitatively similar trend with the analytic theories. All these results support our argument that a reduced $J_{ex}$ at the hotter region of FM (= FM2) enhances $V_{SSE}$.

To experimentally demonstrate the enhancement of $V_{SSE}$ by reducing $J_{ex}$ at the hotter region, we employ a W (4 nm)/CoFeB (2 nm)/AlO$_x$ (2 nm) wire device, in which a ZrO$_2$ gate oxide and a Ru gate electrode are incorporated (Fig. 2a; see details in the Methods section). In this structure, we reduce $J_{ex}$ at the hotter region of CoFeB by utilizing gate voltage ($V_G$)-induced oxygen migration, which is known to efficiently modulate magnetic properties of FM through redox reactions[23,24]. $V_G$-induced oxygen migration allows us to control the oxidation state (i.e., magnetic properties) of FM in *single* sample and thus to avoid possible ambiguities caused by sample-to-sample variations.

To confirm that $V_G$ modulates $J_{ex}$ of CoFeB, we measure temperature-dependent anomalous Hall resistance ($R_H$) of a W (4 nm)/CoFeB (1 nm)/AlO$_x$ (2 nm) sample with $V_G = \pm13$ V (equivalent to 3.25 MV/cm). Here $V_G$ is applied for 5 min at 100 °C, and the measurement is carried out with the gate floating. This is possible due to the non-volatile nature of the gate effect in our sample, which persists even after $V_G$ is turned off. As CoFeB (1 nm) has the in-plane magnetization, we apply an out-of-plane field of 9 T to measure $R_H$. This field is much larger than the demagnetization field (~ 1 T) so that $V_G$-induced change in the magnetic anisotropy does not affect $R_H$ versus temperature. Figure 2b shows that the normalized $R_H$ gradually decreases with temperature, and in particular the sample with $V_G = -13$ V has a stronger temperature dependence. Since the magnetic moment of the CoFeB does not change at the measurement temperature up to 380 K (Supplementary Note 3), this result shows that the Curie temperature and $J_{ex}$ of CoFeB decrease with a negative voltage. Note that in our sign convention, oxygen ions are drifted toward (away from) the CoFeB/AlO$_x$ interface for a negative (positive) $V_G$. Therefore, the result of Fig. 2b indicates that the oxidation reduces $J_{ex}$ of CoFeB near the CoFeB/AlO$_x$ interface.

We next measure a thermoelectric voltage ($V_{TE}$) at various $V_G$. In the sample, a vertical temperature gradient ($\nabla T_z$) is generated by irradiating a focused laser (laser power = 30 mW) with a wavelength of 660 nm, and then $V_{TE}$ is measured while sweeping a magnetic field ($B_x$) in the direction transverse with the voltage probe or rotating a magnetic field of 100 mT in the x-y plane (azimuthal angle $\varphi_B$). Figure 2c shows the results; for the case of without applying $V_G$ (black symbols), $V_{TE}$ is positive for a positive $B_x$ (i.e., $M$//+x where $M$ is the CoFeB magnetization), and changes its sign when $M$ is reversed. Defining $\Delta V_{TE} = [V_{TE} (M$//+x$) - V_{TE} (M$//-x$)]/2$, we find that $\Delta V_{TE}$ is 0.7 μV for the case without $V_G$ but is largely modulated by applying $V_G$; $\Delta V_{TE}$ is 5.5 μV for $V_G = +13$ V, which is almost eight times greater than that without $V_G$. Furthermore, $\Delta V_{TE}$ even changes the sign by a negative $V_G$; $\Delta V_{TE} = -4.3$ μV for $V_G = -13$ V.

The $V_G$-induced modulation of $\Delta V_{TE}$ is further confirmed by the angle-dependent $V_{TE}$ measurement that exhibits $\cos\varphi_B$−dependence (Fig. 2d). This angular dependence is consistent with the symmetry of spin thermoelectrics[5] where $V_{TE}$ maximizes when $M$ is aligned in the x-direction perpendicular to both $\nabla T_z$ and the voltage probes. We also

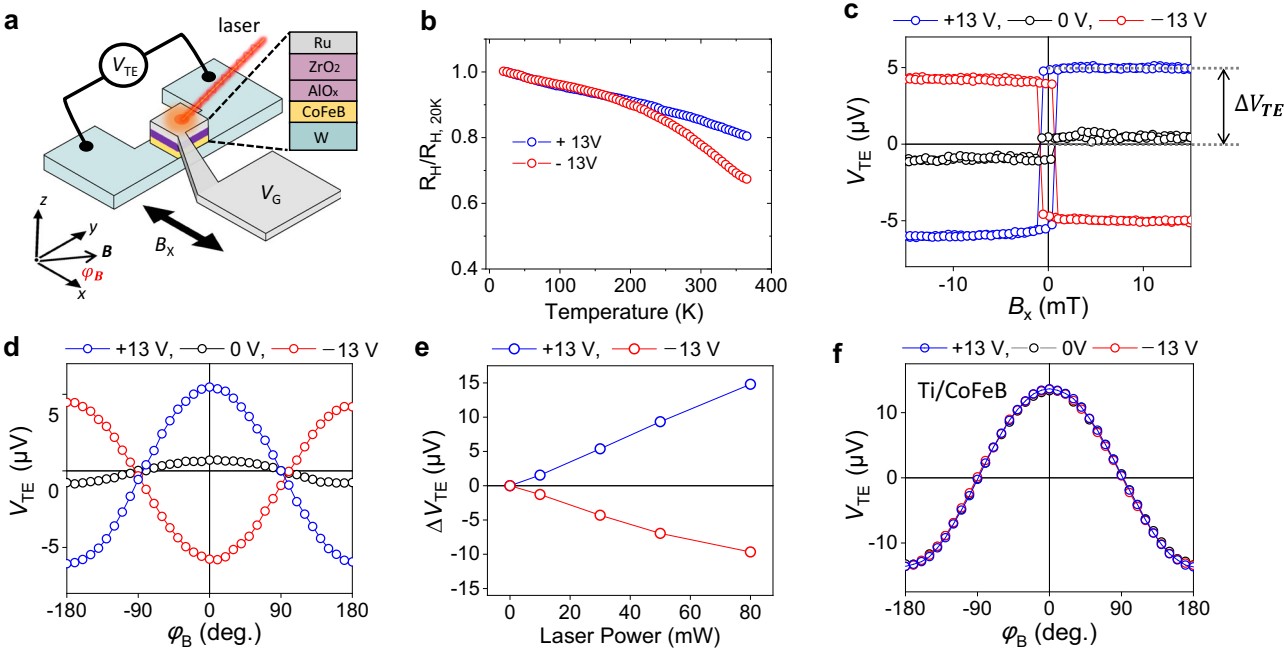

**Fig. 2 | Gate voltage control of thermoelectric voltage. a** Schematic illustration of a gate voltage ($V_G$) modulation of transverse thermoelectric voltage ($V_{TE}$) in a W/CoFeB/AlO$_x$ device with a ZrO$_2$ gate oxide and a Ru gate electrode. **b** Normalized $R_H$ versus temperature in a W (4 nm)/CoFeB (1 nm)/AlO$_x$ (2 nm) structure for $V_G = +13$ V (blue) and $V_G = -13$ V (red). **c** $V_{TE}$ in a W (4 nm)/CoFeB (2 nm)/AlO$_x$ (2 nm) structure as a function of magnetic field ($B_x$) and (**d**), magnetic field angle ($\varphi_B$) at $V_G = +13$ V (blue), −13 V (red), and 0 V (black). **e** Laser power dependence of $\Delta V_{TE}$. **f** $V_G$ effect on $V_{TE}$ in a Ti (3 nm)/CoFeB (2 nm)/AlO$_x$ (2 nm) structure for $V_G = +13$ V (blue), −13 V (red), and 0 V (black). Here, all measurements were done at a laser power of 30 mW except for the experiment of $\Delta V_{TE}$ versus laser power shown in (**e**).

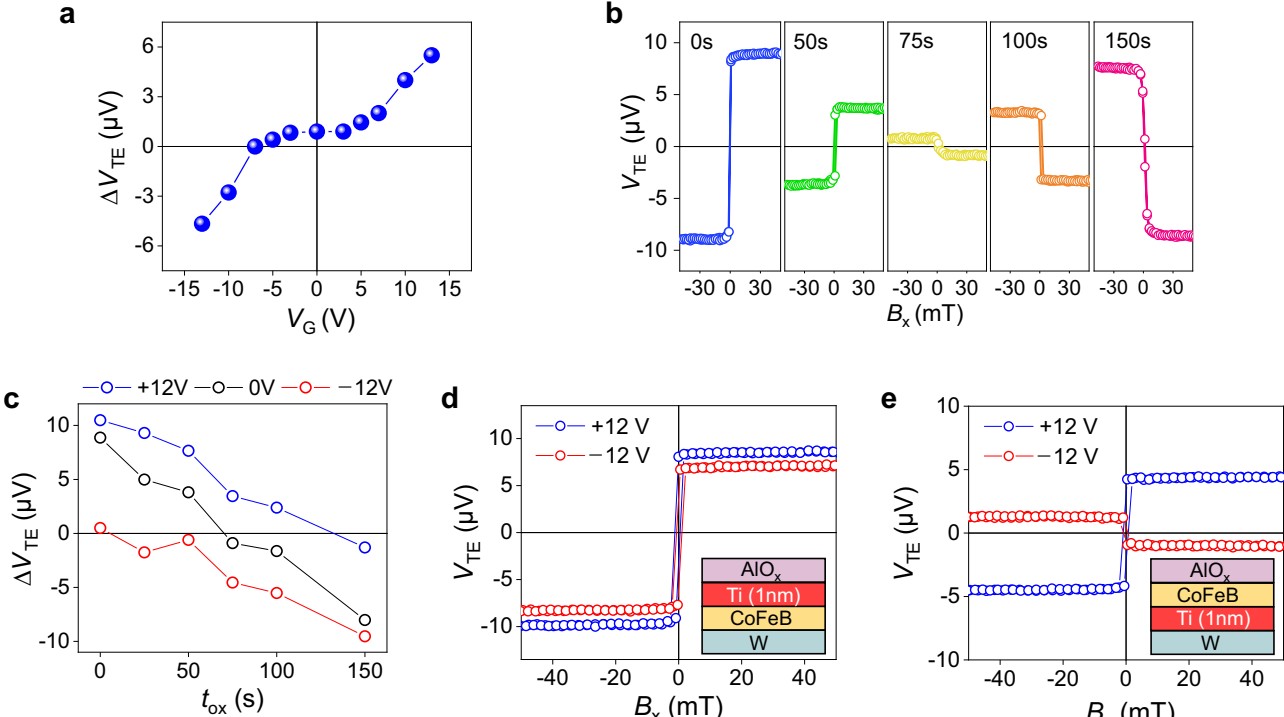

**Fig. 3 | Effect of CoFeB oxidation on $V_{TE}$ in W/CoFeB/AlO$_x$ structure. a** $\triangle V_{TE}$ as a function of $V_G$ in W (4 nm)/CoFeB (2 nm)/AlO$_x$ (2 nm) (laser power = 30 mW). **b** $V_{TE}$ as a function of magnetic field ($B_x$) in W (4 nm)/CoFeB (2 nm)/AlO$_x$ (1.5 nm) with varying plasma oxidation time from 0s to 150s. **c** $\Delta V_{TE}$ as a function of plasma oxidation time $t_{ox}$ with $V_G = 0$ V (black), $V_G = +12$ V (blue) and $-12$ V (red). **d, e** $V_{TE}$ as a function of magnetic field ($B_x$) in a W (4 nm)/CoFeB (2 nm)/Ti (1 nm)/AlO$_x$ (2 nm) structure (**d**) and a W (4 nm)/Ti (1 nm)/CoFeB (2 nm)/AlO$_x$ (2 nm) structure (**e**) with $V_G = +12$ V (blue) and $-12$ V (red) (laser power = 30 mW).

measure $\Delta V_{TE}$ as a function of laser power to confirm that $\Delta V_{TE}$ originates from $\nabla T_z$. Figure 2e shows that $\Delta V_{TE}$ increases linearly with the laser power at both $V_G$ polarities, demonstrating that the $V_G$-induced $\Delta V_{TE}$ enhances by increasing $\nabla T_z$ in the sample.

All the above results are consistent with spin thermoelectric voltages induced by $\nabla T_z$, but $V_G$-induced sign change of $\Delta V_{TE}$ demands a further investigation. This sign change is caused by the fact that CoFeB is a metallic FM and thus not only SSE of W/CoFeB but also anomalous Nernst effect (ANE) of CoFeB itself contributes to $\Delta V_{TE}$. For a W/CoFeB bilayer where W has a negative spin Hall angle[25], the ANE of CoFeB has the opposite sign to the SSE of W/CoFeB[26]. Then, an important question is whether the ANE or the SSE changes with $V_G$. To address this question, we examine $V_G$-induced $\Delta V_{TE}$ in a Ti (3 nm)/CoFeB (2 nm)/AlO$_x$ (2 nm) sample. It is expected that the SSE contribution to $\Delta V_{TE}$ is negligible in this sample because of a negligibly small spin Hall angle of Ti[27]. Figure 2f shows that the angle-dependent $V_{TE}$'s of Ti/CoFeB sample at different $V_G$'s are almost identical regardless of $V_G$. Since $V_{TE}$ of Ti/CoFeB sample is predominantly determined by the ANE of CoFeB, this result shows that the effect of $V_G$ on ANE is negligible. Therefore, the $V_G$-induced $\Delta V_{TE}$ in the W/CoFeB sample is almost entirely governed by the $V_G$-induced modulation of SSE and a negative $V_G$ increases the magnitude of the SSE since the sign of SSE is negative for the W/CoFeB sample. This $V_G$-induced sign change of $\Delta V_{TE}$ in the W/CoFeB sample evidences that the $V_G$-induced modulation of SSE signal is sufficiently large to exceed the ANE signal that is almost independent of $V_G$. We also investigate samples with other NM electrodes of Pt or Ta (Supplementary Note 4). Both samples show that the magnitude of the SSE increases at negative $V_G$, consistent with the results of the W/ CoFeB sample. Furthermore, the $V_G$ modulation efficiency is closely related to the magnitude and sign of the spin Hall angle of the NM layer.

The above results unambiguously show that the gate voltage modifies the SSE. We attribute the main mechanism of the gate effect

to the oxygen ion migration and associated oxidation of CoFeB because it exhibits non-volatile nature and has a threshold voltage. Figure 3a shows that $\Delta V_{TE}$ does not change significantly for $|V_G| < 5$ V. This threshold behavior may be related to the energy barrier for oxygen migration[23,24]. On the other hand, when $V_G$ exceeds the threshold, $\Delta V_{TE}$ increases with $V_G$. This result demonstrates that $\Delta V_{TE}$ can be continuously controlled by $V_G$ in a reversible manner.

To independently confirm that the observed gate effect originates from the oxidation of CoFeB, we directly oxidize the CoFeB layer by plasma oxidation. We fabricate W (4 nm)/CoFeB (2 nm)/AlO$_x$ (1.5 nm) samples where the oxidation state of CoFeB is modulated by varying the oxidation time ($t_{ox}$), which ranges from 0 to 150 s. Figure 3b shows that $V_{TE}$ strongly depends on $t_{ox}$; $\Delta V_{TE}$ is positive for $t_{ox} = 0$ s, gradually decreases with $t_{ox}$, and becomes negative when $t_{ox}$ exceeds 75 s. It indicates that the magnitude of SSE increases with $t_{ox}$, which is the same trend as applying a negative $V_G$ (Fig. 2). This result can be understood by the fact that a negative $V_G$ makes oxygen ions to migrate from oxide to CoFeB, leading to the oxidation of CoFeB. Note that the oxidation-induced modulation of $\Delta V_{TE}$ is also seen in samples with thicker CoFeB (Supplementary Note 5).

To prove that the gate effect is equivalent to the CoFeB oxidation, we also investigate the effect of gate voltage ($V_G = \pm12$ V) on $\Delta V_{TE}$ in the samples with different $t_{ox}$'s (Fig. 3c). When applying $V_G = -12$ V (+12 V), $\Delta V_{TE}$ becomes more negative (positive), i.e., SSE increases (decreases) for all samples regardless of $t_{ox}$. These results show that the gate effect is equivalent to the CoFeB oxidation. We also check whether the CoFeB oxidation near the CoFeB/AlO$_x$ interface (i.e., the hotter region) or the CoFeB/W interface (i.e., the colder region) is important for the SSE change. To check this, the interfaces are intentionally modified by introducing a Ti (1 nm) insertion layer. Figure 3d,e show that the $V_G$ effect disappears when the Ti layer is inserted at the CoFeB/AlO$_x$ interface whereas an apparent $V_G$ effect is still present when the Ti layer is inserted at the W/CoFeB interface. This result confirms that the

oxidation state of CoFeB at the CoFeB/AlO$_x$ interface, i.e., the hotter region, is crucial for the SSE. Note that oxidation of the CoFeB layer can modulate the shunting effect and consequently alter the SSE and ANE effects. However, the change in shunting estimated from the resistance change by $V_G$ is only 3.2% (Supplementary Note 6), which is not sufficient to account for the magnitude and even sign modulation of the $V_{TE}$ shown in Fig. 2c.

In this work, we demonstrate that the SSE signal enhances by gate voltage that modulates the oxidation state of FM. Our theory describes the enhancement of SSE with a reduced exchange interaction at the hotter region of FM.

Possible consequences of the oxidation other than a reduced exchange may include an increased damping, a formation of an anti-ferromagnetic phase such as CoO and FeO$_x$ at the interface[28], a reduced saturation magnetization, or a change of magnetic anisotropy. Therefore, we also check whether these changes can explain our experimental observation. Numerical simulations based on the stochastic LLG equation (Supplementary Note 2) show that the increased damping at the hotter region enhances the SSE signal by <50%, which is much weaker than the effect of reduced exchange. Simulations also show that a formation of an antiferromagnetic phase, a reduced saturation magnetization, or a change of magnetic anisotropy of oxidized FM lattices is unable to describe a largely enhanced SSE due to the oxidation. On the other hand, it should be noted that the reduced electronic and phononic heat conductivity in oxidized FM lattices can induce a larger thermal gradient compared to non-oxidized ferromagnet layers, since insulators typically have lower thermal conductivity than metals. Although the net SSE enhancement induced by oxidation can include a contribution from an increased thermal gradient in general, our analysis suggests that the exchange modulation is dominant for our experiment (Supplementary Note 7). Therefore, we conclude that the reduced exchange interaction at the hotter region is a main cause of the observed efficient $V_G$-induced enhancement of SSE.

Although the magnitude of thermoelectric voltages in our device is still far from practical applications for energy harvesters, our work demonstrating a proof-of-concept for enhancing the SSE through the oxygen manipulation in NM/FM/oxide multilayers could pave an efficient way for further development for the SSE-based thermoelectric devices. As we explained in the beginning of this letter, the thermoelectric conversion via SSE consists of three separate processes and the enhancement of SSE in each process can be multiplied. Therefore, our approach, a reduced exchange interaction at the hotter region, can be combined with FM/NM interface engineering and strong spin-orbit-coupled NM to further enhance the SSE signal. Moreover, our approach is far more general than we demonstrate here. The reduced exchange at the hotter region can be realized in magnetic insulators that are widely used for the SSE studies[2,15–21,29,30]. It is not limited to single layer FM but is also applicable to multiple layers consisting of two or more FMs having different exchange interaction, yielding a wide variation of material combinations. Moreover, the enhanced SSE by the exchange engineering can be combined with thermoelectric effect induced by couplings among magnon, electron, and phonon systems to further improve the thermoelectric signal[31–36]. Optimization of such multilayers based on the concept we report here paves a way to realize practical applications based on the SSE and broadens the scope of material engineering for the SSE-based thermoelectric devices.

## Methods

### Sample preparation
Samples of W(or Ti)/CoFeB/AlO$_x$ structure were grown on thermally oxidized Si substrates by magnetron sputtering with a base pressure of $4.0 \times 10^{-6}$ Pa. The metallic layers were deposited at a working Ar pressure of 0.4 Pa and a power of 30 W, and the AlO$_x$ layer was formed by plasma oxidation of an Al layer with an O$_2$ pressure of 4 Pa and a power of 30 W. A bar-shaped device of 15 μm × 1000 μm was patterned

by using photolithography and ion milling process. A gate oxide of ZrO$_2$ (40 nm) was grown at 125 °C by plasma-enhanced atomic layer deposition using a TEMAZ [Tetrakis (ethylmethylamido) zirconium] precursor and O$_2$. The gate electrode (15 μm × 20 μm) of Ru (20 nm) was defined at the centre of the bar-shaped device.

### Thermoelectric measurement
Thermoelectric voltage was measured by illuminating a focused laser with a spot size of 5 μm and a wavelength of 660 nm that generated a vertical temperature gradient. During the measurement, the laser spot was positioned at the centre of the devices by monitoring the reflectance of the laser using a photodiode sensor. Prior to the measurement, a gate voltage was applied to the Ru gate electrode for 5 min at 100 °C with a ground connected to the W/CoFeB electrode. Thermoelectric measurements were carried out with the gate floating at room temperature, and each measurement was repeated more than 3 times.

### Numerical simulation
We perform atomistic model simulations with stochastic Landau-Lifshitz-Gilbert (LLG) equation for one-dimensional system consisting of ten layers along the $z$-axis. The system is allowed to have inhomogeneous exchange (including antiferromagnetic exchange) and damping. The stochastic LLG equation for the unit magnetization $\hat{\mathbf{m}}_i$ of the $i$th layer is $\frac{d\hat{\mathbf{m}}_i}{dt} = -\gamma\hat{\mathbf{m}}_i \times (\mathbf{H}_{\text{eff},i} + \mathbf{H}_{\text{th},i}) + \alpha_i\hat{\mathbf{m}}_i \times \frac{d\hat{\mathbf{m}}_i}{dt}$, where $\gamma$ is the gyromagnetic ratio and $\alpha_i$ is the damping parameter of the $i$th layer. The effective field $\mathbf{H}_{\text{eff},i} \left( = \frac{2A_{ex,i}}{M_s}\frac{\partial^2\hat{\mathbf{m}}_i}{\partial z^2} - \frac{2K_h}{M_s}m_z\hat{\mathbf{z}} + H_{\text{ext}}\hat{\mathbf{x}} \right)$ consists of the exchange field, easy-plane anisotropy field, and external magnetic field $H_{\text{ext}}$, where $A_{ex,i}$ is the exchange stiffness of the $i$th layer, $M_s$ is the saturation magnetization, $K_h$ is the easy-plane anisotropy energy. The thermal fluctuation field $\mathbf{H}_{\text{th},i}$ obeys $\langle\mathbf{H}_{\text{th},i}(t)\rangle = 0$ and $\langle\mathbf{H}_{\text{th},i}(t)\mathbf{H}_{\text{th},j}(t')\rangle = \frac{2k_BT_i\alpha_i}{\gamma VM_s}\delta_{ij}\delta(t-t')$, where $k_B$ is the Boltzmann constant, $T_i$ is temperature of the $i$th layer and $V$ is the volume.

Following Ref. 22, we calculate the SSE signal from $j^s_{Tgrad} - j^s_{Tconst}$ where $j^s_{Tgrad}$ is the spin pumping current proportional to the time average $\langle[\hat{\mathbf{m}}_N \times d\hat{\mathbf{m}}_N/dt]_x\rangle$ with $N = 10$ (i.e., the coldest layer) in the presence of temperature gradient whereas $j^s_{Tconst}$ is the spin pumping current at a constant temperature.

## Data availability
The data that support the findings of this study are available from the corresponding author upon reasonable request.

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

## Acknowledgements

This work was supported by support from Samsung Research Funding Center of Samsung Electronics under Project Number SRFC-MA1802-01 and the National Research Foundation of Korea (NRF) (NRF-2022R1A4A1031349). K.-J.L. was supported by the NRF (NRF-2020R1A2C3013302). K.-J.K and B.-G.P were supported by KAIST-funded Global Singularity Research Program for 2021. K.-W.K. was supported by the KIST institutional program.

## Author contributions

B.-G.P. and K.-J.L. planned and supervised the study. J.-M.K. and M.-G.K. fabricated devices and performed thermoelectric measurements with the help of J.-G.C., S.L., J.P., C.V.P., K.-J. K., and J.-R.J. S.-J.K., K.-W.K. and K.-J.L. performed theoretical and numerical studies. K.-J.L., B.-G.P., J.-M.K., S.-J.K., and M.-G.K. wrote the paper.

## Competing interests

The authors declare no competing interests.
