## [Peer Review File · Nature Communications]

Reviewers' Comments:

Reviewer #1:

Remarks to the Author:

Report on manuscript

Authors: Kim et al

Title: Enhanced spin Seebeck effect via oxygen manipulation

This manuscript reports experimental and theoretical studies of the spin Seebeck effect (SSE) in metal/ferromagnet/oxide nanostructures, in which by applying a thermal gradient perpendicular to the sample plane, an electric voltage is detected along the metallic layer. The authors argue that by oxidizing the interface of the ferromagnetic (FM) layer with the oxide, a second FM layer is formed with an exchange parameter smaller than in the original FM. The authors show theoretically that the smaller exchange parameter in the hotter side of the FM results in a larger SSE in both mechanisms that explain the generation of a spin current by a thermal gradient. The manuscript presents experimental data obtained with a structure in which the FM material is CoFeB and that incorporates a gate electrode which confirm the theoretical prediction. By changing the voltage gate the authors manage to control the oxidized layer in the FM and thus the magnitude of the SSE. The results are very interesting, the experiments and theoretical interpretation are presented in detail, and the paper is well written. In my opinion the paper will attract attention of the magnetism and spintronics community and deserves to be published in Nature Communications. The only minor suggestion is that on page 2, the authors should use the nomenclature established in the spin caloritronic literature, as in Ref. [8]. The longitudinal SSE refers to the configuration in which the temperature gradient is applied perpendicular to the plane, not along the plane, while the transverse configuration applies to the thermal gradient in the plane.

Reviewer #2:

Remarks to the Author:

The manuscript 'Enhanced spin Seebeck effect via oxygen manipulation' by Jeong-Mok Kim et al. reports a substantial enhancement of the Spin Seebeck Effect (SSE) by oxidizing a ferromagnet in normal metal/ferromagnet/oxide structures. In W/CoFeB/AlO_x structures the voltage induced interfacial oxidation of CoFeB modifies the SSE and results in the enhancement of thermoelectric signal by an order of magnitude which is the key finding of this manuscript and is a noteworthy result.

The enhancement of the SSE signal is quite convincingly demonstrated by doing the same experiments on samples that are successively oxidized by a gate voltage or by plasma oxidation with prolonged times and the modification of the SSE signal is easily evident in the bare signal. Thus, as an experimentalist I do not doubt on the validity of the clearly presented data.

For explanation of the effect they evoke a model that claims a reduced exchange interaction in the oxidized region which should increase a temperature difference between magnons and electrons in normal metal and or a larger gradient in the magnon chemical potential. If one assumes spin orbit coupling in the non oxidized heavy metal is unaffected and one uses such a macroscopic description of the SSE either of this must happen, if not completely different mechanisms take place. They elaborate these models which are based on a drift diffusion model or magnon temperature model and can reproduce their results based on a reduced exchange interaction in the ferromagnetic layer.

During the calculations they apply the Landau Lifshitz Gilbert (LLG) equation to each atomic layer separately as the full thickness of the CoFeB layer is only 2nm, approximated by 10 magnetic layers.

While this description can describe the measurements, I have the feeling much more must happen in the material. Applying LLG to single atomic layers is questionable and if, not only exchange and damping but also saturation magnetization and anisotropy would change, so rather a quantum mechanical chain of spins must be treated. Also, heat is mainly transported by electrons and phonons and they play no role in this model, while qualitatively the reduced electronic and

phononic heat conductivity in the oxidized structure must lead to a higher temperature gradient over the layers for a given fixed heat flow. So, a microscopic description for such an ultrathin system is needed that couples electrons, phonons and spins. As the full model might not be possible to solve, an approximate model as presented here can be acceptable. This as well as the model derivations should be checked by a theory colleague.

With respect to the claim in the abstract that SSE offers efficient thermoelectric devices I have more severe doubts. Clearly the geometry with a heat gradient perpendicular to the electric field has considerable advantages in device design and was already discussed for Anomalous Nernst Effect (ANE) and SSE devices. In the system here after oxidation the effect is comparable to that of the ANE, but the latter has not found applications yet due to its (too) low efficiency. A further disadvantage of the SSE based devices is their small thickness that limits both the electric conductance and the achievable temperature difference. In this manuscript no attempt has been made to quantify the thermoelectric efficiency. With 80mW of Laser power on a 5µm diameter spot the heat flow is very high $4e9 \text{ W/m}^2$ and with thermal conductivities of order 10 W/mK for metals and 8nm thickness of the metal layers the temperature difference is just order of 3K, which already due to Carnot's law will restrict efficiency strongly, while additional restrictions apply for thermoelectrics involving irreversible processes limiting efficiency further. The electric field generated is of order 2 V/m assuming it is generated at the laser spot only. Using multilayers can greatly improve this, but simplicity of design and cost suffer severely and realistic heat flows in application should be considered. So for a claim of thermoelectric efficiency I would like to see a relevant calculation of input power versus output power. Being 10x more efficient than the non oxidized case is a step in the right direction of course.

If one accepts the simplified model descriptions the text is clearly presented and self consistent. However, on the relevance for thermoelectric applications realistic statements should be made.

Some other relevant work in that research direction should be added. E.g.

Kurokawa et al Scientific Reports (2022) 12:16605; <https://doi.org/10.1038/s41598-022-21200-9>

Pan et al. Nature Materials| VOL 21 (2022) 203

Yang et al., AIP Advances 7, 095017 (2017); <https://doi.org/10.1063/1.5003611>

Reviewer #3:

Remarks to the Author:

In this paper, the authors found that the inverse spin Hall voltage induced by the spin Seebeck effect (SSE) in W/CoFeB junctions is drastically modulated by the oxidation of CoFeB. The oxygen manipulation in the CoFeB films was realized by electric-field-induced oxygen migration and plasma oxidation. The enhanced SSE voltage can be qualitatively explained by a reduced exchange interaction in CoFeB. The observed behaviors are interesting and the experimental results are systematic. This paper is thus worth publishing. However, for further consideration, the following issues need to be addressed.

1) The electrical resistivity of the CoFeB films with various oxidation conditions should be measured systematically as a function of temperature. The difference in the electrical resistivity affects the shunting effect in the W/CoFeB bilayer structures. When the resistivity of CoFeB increases, the SSE voltage in the W layer (anomalous Nernst voltage in the CoFeB layer) should be enhanced (suppressed) due to the modulated shunting effect. Clarifying the behavior of the shunting effect is necessary to quantitatively discuss the observed SSE voltage.

2) For confirming the enhancement of SSE by the oxidation of CoFeB, systematic measurements using Pt/CoFeB junctions with the Pt layer having the opposite spin Hall angle are necessary. Why do the authors focus only on W? Considering the shunting effect contribution discussed in my comment 1), the Pt/CoFeB junctions are better to extract the SSE contribution.

3) The thickness of the CoFeB layer used in this study is very thin (1 nm) for the effective use of the electric-field-induced oxidation. However, the SSE voltage is known to be monotonically enhanced as the thickness of the magnetic layer increases. Can the SSE voltage for the oxidized

CoFeB further be enhanced by increasing the thickness? This can be checked by the samples prepared by the plasma oxidation. If very large SSE voltage was obtained for thicker samples, the impact of this work would be improved.

4) The laser power should be described for all the data.

5) In Fig. 3c, the ΔV_{TE} value for -12V and $t_{ox} = 0s$ is still positive. Why is this result different from the other data?

6) The authors define ΔV_{TE} as $V_{TE}(M//+x) - V_{TE}(M// -x)$. However, $\Delta V_{TE} = (V_{TE}(M//+x) - V_{TE}(M// -x)) / 2$ is more popular. The ΔV_{TE} values should be replaced based on the standard definition.

7) The sentence "This result shows that the Curie temperature and thus J_{ex} of CoFeB decreases with a negative voltage" (lines 122-123) is not convincing because of the absence of the data at high temperatures. If the oxygen migration in CoFeB is temperature-dependent, the estimation of the Curie temperature from the anomalous Hall data may become complicated. The authors should discuss this point quantitatively.

8) The sentence "For the first process, lowering the damping of FM or reducing the thermal conductivity of FM enhances the SSE efficiency by several factors" (lines 50-52) is misleading. The thermal conductivity and "the first process" (i.e., SSE coefficient) are different parameters and can be designed independently.

Dear Reviewer,

We appreciate reviewers' comments and valuable queries, which have helped us improve the clarity and quality of our manuscript. Given below are detailed point-by-point responses to the questions and suggestions. The corresponding modifications are incorporated in the revised manuscript (marked in blue). We believe our manuscript has been improved significantly and now deserves publication in *Nature Communications*.

Yours sincerely,

Profs. Byong-Guk Park and Kyung-Jin Lee on behalf of all co-authors.

Reviewer #1:

This manuscript reports experimental and theoretical studies of the spin Seebeck effect (SSE) in metal/ferromagnet/oxide nanostructures, in which by applying a thermal gradient perpendicular to the sample plane, an electric voltage is detected along the metallic layer. The authors argue that by oxidizing the interface of the ferromagnetic (FM) layer with the oxide, a second FM layer is formed with an exchange parameter smaller than in the original FM. The authors show theoretically that the smaller exchange parameter in the hotter side of the FM results in a larger SSE in both mechanisms that explain the generation of a spin current by a thermal gradient. The manuscript presents experimental data obtained with a structure in which the FM material is CoFeB and that incorporates a gate electrode which confirm the theoretical prediction. By changing the voltage gate the authors manage to control the oxidized layer in the FM and thus the magnitude of the SSE. The results are very interesting, the experiments and theoretical interpretation are presented in detail, and the paper is well written. In my opinion the paper will attract attention of the magnetism and spintronics community and deserves to be published in Nature Communications.

Response) We appreciate the reviewer for the positive comments of “*The results are very interesting, the experiments and theoretical interpretation are presented in detail, and the paper is well written*” and “*In my opinion the paper will attract attention of the magnetism and spintronics community and deserves to be published in Nature Communications.*”

The only minor suggestion is that on page 2, the authors should use the nomenclature established in the spin caloritronic literature, as in Ref. [8]. The longitudinal SSE refers to the configuration in which the temperature gradient is applied perpendicular to the plane, not along the plane, while the transverse configuration applies to the thermal gradient in the plane

Response) We appreciate the reviewer’s comment, which made us realize that the nomenclature in our manuscript may confuse the thermoelectric effect in longitudinal (transverse) geometry with the longitudinal (transverse) SSE. The former, taken from the references [*Nat. Mater.*, **20**, 463–467 (2021); *Energy Environ. Sci.* **14**, 3480–3491 (2021)], is defined by the relative direction between the applied temperature gradient (∇T) and the generated thermoelectric voltage (V). According to this definition, the conventional Seebeck effect is a thermoelectric effect in longitudinal geometry while both longitudinal and transverse

SSE are thermoelectric effects in transverse geometry (**Figure R1**). So, we classify SSE as a transverse thermoelectric device in the main text. As the reviewer pointed out, this work employs longitudinal SSE, where the thermal spin current is generated along perpendicular ∇T . (**Fig. R1b**). This is distinct from transverse SSE (**Fig. R1c**).

In revised manuscript, we modified sentences on page 2 to avoid any confusion as follows;

“This limitation can be overcome by a thermoelectric device in transverse geometry using the spin Seebeck effect (SSE) that generates an electric voltage in the direction perpendicular to a temperature gradient.”

“the thermoelectric conversion via longitudinal SSE consists of three separate processes”

Figure R1. Thermoelectric effects depending on the geometry of temperature gradient (∇T) and consequent thermoelectric voltage generation (V). **a**, Conventional Seebeck effect in longitudinal geometry. **b,c**, Thermoelectric effects in transverse geometry; longitudinal spin Seebeck effect (**b**) and transverse spin Seebeck effect (**c**). Here, M , and J_s stand for the magnetization, and thermal spin current, respectively, and the arrow indicates the direction of each variables.

Reviewer #2 (Remarks to the Author):

The manuscript 'Enhanced spin Seebeck effect via oxygen manipulation' by Jeong-Mok Kim et al. reports a substantial enhancement of the Spin Seebeck Effect (SSE) by oxidizing a ferromagnet in normal metal/ferromagnet/oxide structures. In W/CoFeB/AlO_x structures the voltage induced interfacial oxidation of CoFeB modifies the SSE and results in the enhancement of thermoelectric signal by an order of magnitude which is the key finding of this manuscript and is a noteworthy result. The enhancement of the SSE signal is quite convincingly demonstrated by doing the same experiments on samples that are successively oxidized by a gate voltage or by plasma oxidation with prolonged times and the modification of the SSE signal is easily evident in the bare signal. Thus, as an experimentalist I do not doubt on the validity of the clearly presented data.

For explanation of the effect they evoke a model that claims a reduced exchange interaction in the oxidized region which should increase a temperature difference between magnons and electrons in normal metal and or a larger gradient in the magnon chemical potential. If one assumes spin orbit coupling in the non oxidized heavy metal is unaffected and one uses such a macroscopic description of the SSE either of this must happen, if not completely different mechanisms take place. They elaborate these models which are based on a drift diffusion model or magnon temperature model and can reproduce their results based on a reduced exchange interaction in the ferromagnetic layer.

Response) We appreciate the reviewer's comment, "*Thus, as an experimentalist I do not doubt on the validity of the clearly presented data.*" Please find our responses to the comments below, which has been reflected in the revised manuscript. This hopefully alleviates the reviewer's concerns and makes revised manuscript acceptable for publication.

During the calculations they apply the Landau Lifshitz Gilbert (LLG) equation to each atomic layer separately as the full thickness of the CoFeB layer is only 2nm, approximated by 10 magnetic layers. While this description can describe the measurements, I have the feeling much more must happen in the material. Applying LLG to single atomic layers is questionable and if, not only exchange and damping but also saturation magnetization and anisotropy would change, so rather a quantum mechanical chain of spins must be treated.

Response) We thank the reviewer for this comment. Concerning the applicability of LLG to single atomic layers and the necessity of quantum mechanical treatment of spins, it is noted that the (classical) atomistic LLG in the presence of thermal fluctuation is widely used in the community and can reasonably describe experimental results [e.g., Evans, R. F. L. et al., J. Phys.: Condens. Matter 26 (2014) 103202]. In this respect, we believe that our atomistic LLG simulation provides insight into which parameters can be manipulated to enhance SSE. We also emphasize that our simulation result is in good qualitative agreement with analytic theories.

In response to the comment about possible changes of the anisotropy and the saturation magnetization due to oxidation, we perform LLG simulation with varying one of these two magnetic properties while fixing other properties (**Fig. R2**). In this simulation, non-oxidized FM lattices have fixed easy-plane anisotropy energy ($K_{h,0} = 4 \times 10^6 \text{ erg cm}^{-3}$) and saturation magnetization ($M_{s,0} = 1000 \text{ emu cm}^{-3}$), while we vary these parameters in oxidized FM lattices as $-2 K_{h,0} \leq K_{h,ox} \leq 2K_{h,0}$ for Fig. R2a and $0.05M_{s,0} \leq M_{s,ox} \leq M_{s,0}$ for Fig. R2b. In **Fig. R2a**, a negative $K_{h,ox}$ means a perpendicular anisotropy of the oxidized FM lattices, assuming interfacial perpendicular anisotropy induced by an adjacent oxide layer. In **Fig. R2b**, we consider the cases with $M_{s,ox}$ smaller than $M_{s,0}$ since the oxidation of magnetic materials induces a reduction of the Curie temperature, resulting in a reduction of the saturation magnetization.

Figure R2a shows the result with various $K_{h,ox}$ in oxidized FM lattices. The spin pumping current is found to change within 10 %, compared to that of the homogeneous anisotropy ($K_{h,ox} = K_{h,0}$) case. Therefore, the anisotropy change due to oxidation has no noticeable effect on the SSE. This tendency can be understood by the dominance of exchange interaction over inhomogeneous anisotropy energy in thermal spin pumping. **Figure R2b** shows the result with various $M_{s,ox}$ in oxidized FM lattices. The simulation shows that the spin pumping current is reduced when the oxidized FM lattices have a smaller saturation magnetization than non-oxidized FM lattices. It is a contrary tendency from the experiment. We understand this tendency as a consequence of an enhanced effective exchange field ($H_{ex} = 2A_{ex}/M_s$) in the oxidized FM lattices. Therefore, varying the anisotropy energy or the saturation magnetization of oxidized FM lattices is unable to describe a largely enhanced SSE due to the oxidation.

We added the above discussion on page 10 and related discussion in Supplementary Note 2 as;

“Possible consequences of the oxidation other than a reduced exchange may include an increased damping, a formation of an antiferromagnetic phase such as CoO and FeO_x at the interface, a reduced saturation magnetization, or a change of magnetic anisotropy. Therefore, we also check whether these changes can explain our experimental observation”

“Simulations also show that a formation of an antiferromagnetic phase, a reduced saturation magnetization, or a change of magnetic anisotropy of oxidized FM lattices is unable to describe a largely enhanced SSE due to the oxidation”

Figure R2. The ratio $J_{s,inhomo}/J_{s,homo}$ with varying **a**, anisotropy energy and **b**, the saturation magnetization in oxidized FM lattices. Parameters: $M_{s,0} = 1000 \text{ emu cm}^{-3}$, $K_{h,0} = 4 \times 10^6 \text{ erg cm}^{-3}$, $\alpha_i = 0.01$, $A_{ex,i} = 2 \times 10^{-7} \text{ erg cm}^{-1}$, $H_{ext} = 100 \text{ Oe}$, and the unit cell volume = $1000 \times 1000 \times 0.4 \text{ nm}^3$.

Also, heat is mainly transported by electrons and phonons and they play no role in this model, while qualitatively the reduced electronic and phononic heat conductivity in the oxidized structure must lead to a higher temperature gradient over the layers for a given fixed heat flow.

Response) We thank the reviewer for this comment that we have ignored in the interpretation of previous manuscript. As noted by the reviewer, the contribution of heat transfer from electrons and phonons cannot be ignored, particularly given that insulators typically have lower thermal conductivity compared to metals. This results in a larger thermal gradient for oxidized FM lattices compared to non-oxidized ones. Consequently, the FM oxidation leads to an

increase in total SSE not only due to the exchange modulation but also due to the enhanced thermal gradient.

In order to find out which one between the exchange modulation and the enhanced thermal gradient is dominant for the total SSE signal of our experiment, we carry out the following analysis. First, we estimate the SSE enhancement by oxidation from experimental results. Both ANE voltage V_{ANE} and SSE voltage V_{SSE} contribute to net thermoelectric voltage V_{TE} . They are related by a parallel circuit model [Adv. Funct. Mater. **26**, 5507-5514 (2016)];

$$\frac{V_{TE}}{R} = \frac{V_{ANE}}{R_F} + \frac{V_{SSE}}{R_N}, \quad (R1)$$

where $R [= 1/(R_F^{-1} + R_N^{-1})]$ is the total resistance of FM/normal metal (NM) bilayer and R_F (R_N) is the resistance of FM (NM).

We estimate V_{ANE} from V_{TE} data of Ti (3 nm)/CoFeB (2 nm)/AlO_x sample assuming no SSE contribution to V_{TE} . It is because this sample has a negligible V_{SSE} [i.e., a small spin Hall angle of Ti and almost no change in V_{TE} at the gate voltage of ± 13 V (see Fig. 2f of main text)]. The resistivity of CoFeB is measured to be $160 \mu\Omega \cdot \text{cm}$ (**Fig. R3**), which gives $R_{CoFeB} = 53.3 \text{ k}\Omega$ (device width $w = 15 \mu\text{m}$ and device length $L = 1,000 \mu\text{m}$). From the measured total resistance ($R_{Ti/CoFeB} = 38.0 \text{ k}\Omega$), we obtain $R_{Ti} = 132.2 \text{ k}\Omega$. Using Eq. (R1) and the measured V_{TE} ($= 13.7 \mu\text{V}$; Fig. 2f of main text), we then obtain V_{ANE} of $19.2 \mu\text{V}$.

Figure R3. $R^{-1}L/w$ as a function of t_{CoFeB} (device width $w = 15 \mu\text{m}$, device length $L = 1,000 \mu\text{m}$). (device width $w = 5 \mu\text{m}$, device length $L = 35 \mu\text{m}$).

The gate voltage (V_G) induced SSE enhancement of W (4 nm)/CoFeB (2 nm)/AlO_x sample is estimated as follows. R_W is calculated to be 32.0 k Ω using the measured total resistance ($R_{W/CoFeB} = 20.0$ k Ω) and $R_{CoFeB} = 53.3$ k Ω . Using Eq. (R1) and V_{TE} of W/CoFeB/AlO_x sample with $V_G = +13$ V (Fig. 2c of main text), we obtain V_{SSE} at $V_G = +13$ V to be -2.7 μ V. On the other hand, using the same procedure, we obtain V_{SSE} at $V_G = -13$ V to be -18.4 μ V, which is enhanced in magnitude by oxidation. Then, the SSE enhancement ratio by oxidation is about 5.8 [= (18.4-2.7)/2.7]. It is noted that we do not consider the oxidation induced reduction in the CoFeB thickness for this estimation. When we consider this reduction, the SSE enhancement ratio becomes 5.3. Therefore, the oxidation of CoFeB results in the SSE enhancement by about five times.

Next, we estimate the SSE enhancement by an increased thermal gradient of oxidized FM lattices. By comparing the resistances of W-based sample at $V_G = \pm 13$ V (**Fig. R3b**; device width $w = 5$ μ m, device length $L = 35$ μ m), we estimate the thickness of oxidized CoFeB to be 0.17 nm. We use the heat transfer module of the COMSOL software to calculate the temperature profile of Ru(20)/ZrO₂(40)/AlO_x(2)/CoFeB(2)/W(4)/SiO₂(100)/SiO in **Fig. R4a** and Ru(20)/ZrO₂(40)/AlO_x(2)/CoFeB-oxidized(0.17)/CoFeB(1.83)/W(4)/SiO₂(100)/SiO in **Fig. R4b**, where the numbers in parentheses are the thicknesses in the unit of nm. We use parameters of CoO_x for the CoFeB-oxidized layer because we can find all parameters necessary for COMSOL simulation for CoO_x. A monochromatic 5- μ m continuous laser beam of 30 mW is applied on the surface of Ru as in our experiment.

From the COMSOL simulation, the thermal gradient of non-oxidized FM (CoFeB) layer is 1.91 K/ μ m (**Fig. R4a**), while the thermal gradient of oxidized FM (CoO_x) layer is 8.31 K/ μ m (**Fig. R4b**). Then the weighted average of thermal gradient for the CoO_x (0.17 nm)/CoFeB (1.83 nm) bilayer is 2.45 K/ μ m [= (8.31 \times 0.17+1.91 \times 1.83)/2], which gives an increase of thermal gradient by 28 % [= (2.45/1.91-1) \times 100], as compared to un-oxidized CoFeB. This value (= 28 %) is far smaller than the net SSE enhancement (> 500 %) estimated above.

We also check a possibility of the formation of oxides (Fe₂O₃, FeO, Fe₃O₄) other than CoO_x. For these Fe-based oxides, we cannot find all parameters necessary for COMSOL simulations so that we estimate the thermal gradient assuming the inverse proportionality of the thermal gradient to the thermal conductivity as this assumption is consistent with our COMSOL simulation [thermal conductivity of CoFeB (κ_{CoFeB}) = 87 W/(m K)⁻¹, thermal conductivity of

CoO_x (κ_{CoO_x}) = 20 W/(m K)⁻¹, calculated thermal gradient of CoFeB (dT_{CoFeB}) = 1.91 K/ μm , and calculated thermal gradient of CoO_x (dT_{CoO_x}) = 8.31 K/ μm ; $\frac{\kappa_{\text{CoFeB}}}{\kappa_{\text{CoO}_x}} \approx \frac{dT_{\text{CoO}_x}}{dT_{\text{CoFeB}}}$]. From literature [Takeda, M *et al.*, *Mater. Trans.* **50**, 2242-2246 (2009)], we find $\kappa_{\text{Fe}_2\text{O}_3} \approx 15$ W/(m K)⁻¹, $\kappa_{\text{FeO}} \approx 10$ W/(m K)⁻¹, and $\kappa_{\text{Fe}_3\text{O}_4} \approx 6$ W/(m K)⁻¹. Using the inverse proportionality, these values gives $dT_{\text{Fe}_2\text{O}_3} \approx 11.1$ K/ μm , $dT_{\text{FeO}} \approx 16.6$ K/ μm , and $dT_{\text{Fe}_3\text{O}_4} \approx 27.7$ K/ μm , respectively. Then the values of weighted average of thermal gradient are increased by 41 % for Fe_2O_3 , 65 % for FeO, and 115 % for Fe_3O_4 , respectively. Overall, therefore, the enhancement of SSE due to the increased thermal gradient of oxides is insufficient to describe the net SSE enhancement (> 500 %) estimated from our measurement. This analysis suggests that the exchange modulation is more dominant for the observed SSE enhancement by oxidation than the increased thermal conductivity.

We added the above discussion on page 10 and Supplementary Note 7 as;

“On the other hand, it should be noted that the reduced electronic and phononic heat conductivity in oxidized FM lattices can induce a larger thermal gradient compared to non-oxidized ferromagnet layers, since insulators typically have lower thermal conductivity than metals. Although the net SSE enhancement induced by oxidation can include a contribution from an increased thermal gradient in general, our analysis suggests that the exchange modulation is dominant for our experiment (Supplementary Note 7).”

Figure R4. Vertical temperature profiles calculated from COMSOL simulation for a, Ru(20 nm)/ZrO₂(40 nm)/AlO_x(2 nm)/CoFeB(2 nm)/W(4 nm)/SiO₂(100 nm)/SiO and b, Ru(20 nm)/ZrO₂(40 nm)/AlO_x(2 nm)/CoO_x(0.17 nm)/CoFeB(1.83 nm)/W(4 nm)/SiO₂(100 nm)/SiO.

nm)/ZrO₂(40 nm)/AlO_x(2 nm)/CoO_x(0.17 nm)/CoFeB(1.83 nm)/W(4 nm)/SiO₂(100 nm)/SiO samples. Parameters for COMSOL simulation are as follows: Length of sample: 1 mm, width of sample: 15 μm, thickness of substrate= 649.9 μm, laser spot radius = 5 μm, laser power = 30 mW, and total reflectance of sample = 0.77. Thermal conductivities of CoO_x, CoFeB, and AlO_x are 20, 87 and 3.3 W/(m K)⁻¹, respectively. Thermal conductivity of CoO_x is taken from Lewis, F. B. & Saunders, N. H. *J. Phys. C: Solid State Phys.* **6**, 2525 (1973).

So, a microscopic description for such an ultrathin system is needed that couples electrons, phonons and spins. As the full model might not be possible to solve, an approximate model as presented here can be acceptable. This as well as the model derivations should be checked by a theory colleague.

Response) We agree with the reviewer's remark, '*microscopic description for such an ultrathin system is needed that couples electrons, phonons, and spins*' for accurate description. We below discuss how these couplings affect our experiment.

Magnon-electron coupling can induce magnon-drag thermopower, which contributes to the anomalous Nernst effect (ANE) [S. J. Watzman *et al.*, *Phys. Rev. B* **94**, 144407 (2016)]. However, we have confirmed that there is no noticeable enhancement of ANE by oxidation (Fig. 2f of main text).

Magnon-phonon coupling can induce various thermoelectric effects. First, the Nernst effect via magnetostriction can be induced [Yang *et al.*, *AIP Advances* **7**, 095017 (2017)], but it can be considered negligible in the same way as ANE. Second, additional spin Seebeck effect (SSE) via phonon-magnon drag can be induced due to the temperature difference between the magnon temperature in the ferromagnetic film and the phonon temperature of the substrate [Jaworski *et al.*, *PRL* **106**, 186601 (2011)]. However, for apparent phonon-magnon drag, the 'transverse SSE configuration' is required, while our configuration is 'longitudinal SSE'. Third, additional SSE from the thermal gradient of the insulator can be induced [K. Uchida *et al.*, *Nat. Mater.* **10**, 737-741 (2011)]. However, like phonon-magnon drag, the 'transverse SSE configuration' is required for apparent acoustic SSE.

As described above, we surveyed various couplings which could contribute to additional thermoelectric voltage in our research. We show that all the couplings may not be main

contributions to our research results. On the other hand, we could explain the SSE enhancement with magnon-based theories and calculations in the manuscript, which gives straight insight to the readers. We expect that the detailed SSE study considering all the couplings would be beyond the scope of our study.

We modified a sentence on page 11 as;

“Moreover, the enhanced SSE by the exchange engineering can be combined with thermoelectric effect induced by couplings among magnon, electron, and phonon systems to further improve the thermoelectric signal³¹⁻³⁶.”

With respect to the claim in the abstract that SSE offers efficient thermoelectric devices I have more severe doubts. Clearly the geometry with a heat gradient perpendicular to the electric field has considerable advantages in device design and was already discussed for Anomalous Nernst Effect (ANE) and SSE devices. In the system here after oxidation the effect is comparable to that of the ANE, but the latter has not found applications yet due to its (too) low efficiency. A further disadvantage of the SSE based devices is their small thickness that limits both the electric conductance and the achievable temperature difference. In this manuscript no attempt has been made to quantify the thermoelectric efficiency. With 80mW of Laser power on a 5 μ m diameter spot the heat flow is very high $4e9$ W/m² and with thermal conductivities of order 10W/mK for metals and 8nm thickness of the metal layers the temperature difference is just order of 3K, which already due to Carnot's law will restrict efficiency strongly, while additional restrictions apply for thermoelectrics involving irreversible processes limiting efficiency further. The electric field generated is of order 2V/m assuming it is generated at the laser spot only. Using multilayers can greatly improve this, but simplicity of design and cost suffer severely and realistic heat flows in application should be considered. So for a claim of thermoelectric efficiency I would like to see a relevant calculation of input power versus output power. Being 10x more efficient than the non-oxidized case is a step in the right direction of course. If one accepts the simplified model descriptions the text is clearly presented and self consistent. However, on the relevance for thermoelectric applications realistic statements should be made.

Response) We thank the reviewer for the valuable comment. We agree that the magnitude of thermoelectric voltages in our device is currently insufficient for an efficient energy harvester; however, as acknowledged by the reviewer, our work demonstrating a proof-of-concept for enhancing the SSE through the oxygen manipulation in NM/FM/oxide multilayers could pave the new way for further development for the SSE-based thermoelectric devices. Note that the thermoelectric conversion via SSE consists of three separate processes: magnon spin current generation in FM, thermal spin pumping from FM to NM, and spin-to-charge conversion via inverse spin Hall effect of NM, and the enhancement of SSE in each process can be multiplied. Therefore, our approach, a reduced exchange interaction at the hotter region, can be combined with FM/NM interface engineering and strong spin-orbit-coupled NM to further enhance the SSE signal.

Moreover, our approach is far more general than we demonstrate here because it can be applied in a wide variety of materials such as magnetic insulators [*Science* **285**, 703-706 (1999), *Sci. Adv.* **3**, e1601614 (2017), *Energy Environ. Sci.* **14**, 3480–3491 (2021)] and multiple layers consisting of two or more FMs having different exchange interaction. In addition, the enhanced SSE by the exchange engineering can be combined with conventional Seebeck effect to further improve the thermoelectric signal [*Nat. Mater.* **20**, 463–467 (2021), *Nat. Mater.* **21**, 203–209 (2022)]. Lastly, our results show that not only the magnitude but also the sign of the thermoelectric voltage can be modulated by the gate voltage (V_G), allowing for alternating alignment of the polarity of thermoelectric voltages in adjacent wire elements in the SSE thermopile [*Science* **321**, 1457 (2008), *Nat. Mater.* **11**, 686–689 (2012), *Nat. Commun.* **11**, 2023 (2020)]. This can lead to a scalable signal enhancement in proportional to the number of wire elements. Although the magnitude of thermoelectric voltages in our device is still far from practical application, the concept we report here paves a way for improving SSE and broadens the scope of material engineering for the SSE-based thermoelectric devices.

We modified the following sentences in the abstract and on page 10 of the revised manuscript as;

“SSE offers the potential for efficient thermoelectric devices because the transverse geometry of SSE enables to utilize waste heat from a large-area source by greatly simplifying the device structure”.

“Although the magnitude of thermoelectric voltages in our device is still far from practical applications for energy harvesters, our work demonstrating a proof-of-concept for enhancing the SSE through the oxygen manipulation in NM/FM/oxide multilayers could pave an efficient way for further development for the SSE-based thermoelectric devices.”

Some other relevant work in that research direction should be added. E.g. Kurokawa et al Scientific Reports (2022) 12:16605; <https://doi.org/10.1038/s41598-022-21200-9>, Pan et al. Nature Materials| VOL 21 (2022) 203, Yang et al., AIP Advances 7, 095017 (2017); <https://doi.org/10.1063/1.5003611>

Response) We feel sorry for the missing relevant references. We have included the papers as references # 30, 32, 33 in the revised manuscript, respectively.

Reviewer #3 (Remarks to the Author):

In this paper, the authors found that the inverse spin Hall voltage induced by the spin Seebeck effect (SSE) in W/CoFeB junctions is drastically modulated by the oxidation of CoFeB. The oxygen manipulation in the CoFeB films was realized by electric-field-induced oxygen migration and plasma oxidation. The enhanced SSE voltage can be qualitatively explained by a reduced exchange interaction in CoFeB. The observed behaviors are interesting and the experimental results are systematic. This paper is thus worth publishing. However, for further consideration, the following issues need to be addressed.

Response) We appreciate the reviewer's comment "*The observed behaviors are interesting and the experimental results are systematic. This paper is thus worth publishing.*" Here, we respond to the reviewer's comments with additional experiments, which hopefully alleviate the reviewer's concerns and make revised manuscript acceptable for publication.

1) The electrical resistivity of the CoFeB films with various oxidation conditions should be measured systematically as a function of temperature. The difference in the electrical resistivity affects the shunting effect in the W/CoFeB bilayer structures. When the resistivity of CoFeB increases, the SSE voltage in the W layer (anomalous Nernst voltage in the CoFeB layer) should be enhanced (suppressed) due to the modulated shunting effect. Clarifying the behavior of the shunting effect is necessary to quantitatively discuss the observed SSE voltage.

Response) We appreciate the reviewer's valuable comment on the shunting effect on the thermoelectric voltage (V_{TE}). We agree that the modulated shunting effect by oxidation of the CoFeB can change V_{TE} by altering spin Seebeck and anomalous Nernst effects in the W/CoFeB/ AlO_x structure. To examine the shunting effect, we first measured the resistivities of the W and CoFeB layers. **Figure R5a** shows the reciprocal of resistance (R) normalized by the length (L) and width (w) of the sample as a function of the CoFeB thickness (t_{CoFeB}) in the W (4 nm)/CoFeB (t_{CoFeB})/ AlO_x (2 nm) structures. From the y -intercept and slope of the linear fit of the graph, the resistivities of W and CoFeB were extracted as $216 \mu\Omega\cdot\text{cm}$ and $160 \mu\Omega\cdot\text{cm}$, respectively. We next measured the temperature dependence of R in the W (4 nm)/CoFeB (1 nm)/ AlO_x (2 nm) structure with different V_G of ± 13 V. **Figure R5b** shows that the sample with $V_G = -13$ V has a larger R than that with $+13$ V, which holds over the entire measurement temperature range from 20 K to 380 K. This indicates that CoFeB is partially oxidized when

negative V_G is applied. Note that the weak temperature dependence of R is attributed to the amorphous characteristic of β tungsten. [Appl. Phys. Lett. 106, 182403 (2015)]. Assuming that the voltage-induced change in R occurs only in the CoFeB layer, we calculated the shunting (S_W) through the W layer of the W(4 nm)/CoFeB (2nm) sample using a parallel circuit model, $S_W = \frac{R_W^{-1}}{R_W^{-1} + R_{\text{CoFeB}}^{-1}}$. Here, $R_{W(\text{CoFeB})}$ is the resistance of the W (CoFeB) layer. S_W value is ~ 0.62 at $V_G = +13$ V and increases to ~ 0.64 at $V_G = -13$ V. As the reviewer pointed out, the increased S_W can increase the SSE (decrease the ANE) in the W/CoFeB sample. However, it is believed that this small change in the shunting effect (increase by $\sim 3.2\%$) is not sufficient to account for the modulation of the magnitude and even sign of the thermoelectric voltage (V_{TE}) by V_G shown in Fig. 2c of the main text. Furthermore, it is shown in Fig. 2f of the main text that V_{TE} is not affected by V_G in the Ti/CoFeB sample. Since a similar shunting change also occurs due to the V_G -induced oxidation in the sample, this result supports that the shunting effect does not significantly modify V_{TE} in our samples.

We included the following sentences regarding the shunting effect on V_{TE} on page 9 of the revised manuscript and Supplementary Note 6 as;

Note that oxidation of the CoFeB layer can modulate the shunting effect and consequently alter the SSE and ANE effects. However, the change in shunting estimated from the resistance change by V_G is only 3.2% (Supplementary Note 6), which is not sufficient to account for the magnitude and even sign modulation of the V_{TE} shown in Fig. 2c.

Figure R5. a, $R^{-1}L/w$ as a function of t_{CoFeB} (device width $w = 15 \mu\text{m}$, device length $L = 1,000 \mu\text{m}$). **b,** Temperature dependence of resistance (R) in W (4 nm)/CoFeB (1 nm)/AlO_x (2 nm) structure with V_G of ± 13 V (device width $w = 5 \mu\text{m}$, device length $L = 35 \mu\text{m}$).

2) For confirming the enhancement of SSE by the oxidation of CoFeB, systematic measurements using Pt/CoFeB junctions with the Pt layer having the opposite spin Hall angle are necessary. Why do the authors focus only on W? Considering the shunting effect contribution discussed in my comment 1), the Pt/CoFeB junctions are better to extract the SSE contribution.

Response) We appreciate the reviewer's comment on additional experiments with a Pt electrode with opposite spin Hall angle. In order to respond to the reviewer's question, we fabricated a Pt (3 nm)/CoFeB (2 nm)/AlO_x (2 nm) structure and examined the V_G -induced ΔV_{TE} . Because of a positive spin Hall angle of Pt, the SSE voltage of this sample is expected to be opposite to that of the sample with W (**Figure R6a**). Using the same conditions as in Fig. 2d of the main text, V_{TE} was measured while rotating a magnetic field of 100 mT in the x - y plane (azimuthal angle φ_B). **Figure R6b** shows that the magnitude of ΔV_{TE} of the Pt/CoFeB sample increases when $V_G = -13$ V, which is different from that of the W/CoFeB samples (Fig. 2d of the main text). Since the SSE has the same sign as the anomalous Nernst effect (ANE) in the Pt/CoFeB sample with a positive spin Hall angle, the increase in ΔV_{TE} indicates that the magnitude of the SSE in the Pt/CoFeB sample increases at a negative V_G . This is consistent with the results of the W/CoFeB sample. We also checked a Ta/CoFeB sample with a negative spin Hall angle of Ta. **Figure R6c** shows the angle-dependent V_{TE} of the Ta/CoFeB sample with $V_G = \pm 13$ V; ΔV_{TE} decreases when a V_G of -13V is applied, the same as for the W/CoFeB sample. Finally, we compare the V_G -induced ΔV_{TE} between the samples using the V_G modulation efficiency $\xi = [\Delta V_{TE}(V_G = +13V) - \Delta V_{TE}(V_G = -13V)]/R$, where R is the sample resistance. **Figure R6d** plots ξ for each sample along with the spin Hall angle of the non-magnetic electrodes [Phys. Rev. Lett. 197201 (2014), Appl. Phys. Lett. 107, 232408 (2015)]. This demonstrates that ξ is closely related to the magnitude and sign of the spin Hall angle. These results corroborate that V_G modulates the SSE in the NM/CoFeB structures.

We included the following sentences regarding the V_G -induced ΔV_{TE} in other NM electrodes of Pt or Ta on page 8 of the revised manuscript and Supplementary Note 4 as;

“We also investigate samples with other NM electrodes of Pt or Ta (Supplementary Note 4). Both samples show that the magnitude of the SSE increases at negative V_G , consistent with the results of the W/CoFeB sample. Furthermore, the V_G modulation efficiency is closely related to the magnitude and sign of the spin Hall angle of the NM layer.”

Figure R6. **a** Schematic of a V_G modulation of thermoelectric voltage (V_{TE}) depending on the NM materials with different spin Hall angle (θ_{SH}) in a NM (3 nm)/CoFeB (2 nm)/AlO_x (2 nm) structure. **b-c**, V_{TE} as a function of magnetic field angle (φ_B) at $V_G = +13 V$ (blue), $-13 V$ (red) for a Pt/CoFeB/AlO_x (b) and Ta/CoFeB/AlO_x (c) structure. **d**, The V_G modulation efficiency (ξ) of NM (3 nm)/CoFeB (2 nm)/AlO_x (2 nm) samples along with spin Hall angle of NM. Here, NM is Ti, Ta, Pt, and W.

3) The thickness of the CoFeB layer used in this study is very thin (1 nm) for the effective use of the electric-field-induced oxidation. However, the SSE voltage is known to be monotonically enhanced as the thickness of the magnetic layer increases. Can the SSE voltage for the oxidized CoFeB further be enhanced by increasing the thickness? This can be checked by the samples prepared by the plasma oxidation. If very large SSE voltage was obtained for thicker samples, the impact of this work would be improved.

Response) We appreciate the reviewer for the suggestion that SSE can be further enhanced by increasing the CoFeB thickness. First of all, we would like to note that the CoFeB of 1 nm was used only to observe the temperature dependence of R_H in Fig. 2b of the main text. In this experiment, we used a thin CoFeB layer to clearly demonstrate the voltage-induced oxidation effect. In all other thermoelectric experiments, 2 nm of CoFeB was used. To check if the SSE voltage increases with CoFeB thickness, we fabricated a sample of a Pt (3 nm)/CoFeB (6 nm)/AlO_x (1.5 nm) structure and measured thermoelectric voltage (V_{TE}). Note that we use a Pt

electrode with a positive spin Hall angle that makes the ANE and SSE have the same sign in the Pt/CoFeB structure, allowing us to directly compare the magnitude of V_{TE} (ΔV_{TE}) between the samples with different CoFeB thicknesses. **Figure R7a** shows V_{TE} of the Pt (3 nm)/CoFeB (6 nm)/AlO_x (1.5 nm) structure while sweeping an in-plane magnetic field (B_x). The ΔV_{TE} value of the sample is $\sim 8.7 \mu\text{V}$, which is much larger than that of the Pt (3 nm)/CoFeB (2 nm)/AlO_x (1.5 nm) structure ($\sim 5.4 \mu\text{V}$) with a similar oxidation time as shown above in Fig. R6. In addition, we found that the ΔV_{TE} magnitude is further enhanced by increasing the plasma oxidation time from 100 s to 400 s (**Fig. R7b**), which is consistent with the results in Fig. 3 of the main text. We confirm that the main result of our study, the enhancement of TE signal by interfacial oxidation, also works for samples with thicker ferromagnetic materials, but, additional studies are needed to quantitatively determine the effect of increasing ferromagnet thickness on the TE.

We included the following sentences regarding ΔV_{TE} in thicker CoFeB on page 9 of the revised manuscript and Supplementary Note 5 as;

“Note that the oxidation-induced modulation of ΔV_{TE} is also seen in samples with thicker CoFeB (Supplementary Note 5).”

Figure R7. a, V_{TE} a function of magnetic field (B_x) in a Pt (3 nm)/CoFeB (2 nm)/AlO_x (1.5 nm) structure with a plasma oxidation of 100 s. **b,** V_{TE} as a function of the plasma oxidation time t_{ox} in Pt (3 nm)/CoFeB (6 nm)/AlO_x (1.5 nm) structures. The blue circle represents a sample with $t_{CoFeB} = 2 \text{ nm}$.

4) The laser power should be described for all the data.

Response) We apologize for the unintentional missing description of the laser power. All experiments in this manuscript were done at a laser power of 30 mW except for the experiment of ΔV_{TE} vs. laser power shown in Fig. 2e. In the revised manuscript, we have included the laser power in figure captions.

5) In Fig. 3c, the ΔV_{TE} value for -12V and $t_{ox} = 0$ s is still positive. Why is this result different from the other data?

Response) We thank the reviewer for this comment on the sign of ΔV_{TE} . Note that V_{TE} of the W/CoFeB sample is determined by both ANE and SSE voltages, where the ANE voltage has a positive sign, opposite to that of SSE. For the sample with $t_{ox} = 0$ s, ΔV_{TE} value without V_G is +8.8 μ V, which is mainly contributed by positive ANE because the magnitude of SSE is small. When a V_G of -12V is applied to the sample, the magnitude of ΔV_{TE} decreases due to the increase in the SSE voltage with a negative sign, but the enhancement of SSE induced by the V_G of the sample with $t_{ox} = 0$ s is not sufficient to overcome the positive-signed ANE voltage. Therefore, the sign of ΔV_{TE} remains positive. This is different from other samples with $t_{ox} > 0$ s, where the ΔV_{TE} is negative when applying $V_G = -12$ V because the magnitude of SSE becomes greater than that of ANE.

6) The authors define ΔV_{TE} as $V_{TE}(M//+x) - V_{TE}(M// -x)$. However, $\Delta V_{TE} = (V_{TE}(M//+x) - V_{TE}(M// -x)) / 2$ is more popular. The ΔV_{TE} values should be replaced based on the standard definition.

Response) We thank for this comment. We have modified the definition of ΔV_{TE} as suggested by the reviewer and applied it to the graphs in Figure 2e and Figure 3a, c of the revised manuscript.

7) The sentence "This result shows that the Curie temperature and thus J_{ex} of CoFeB decreases with a negative voltage" (lines 122-123) is not convincing because of the absence of the data at high temperatures. If the oxygen migration in CoFeB is temperature-dependent, the

estimation of the Curie temperature from the anomalous Hall data may become complicated. The authors should discuss this point quantitatively.

Response) We thank the reviewer for the valuable comment. The reviewer's question is twofold; first, our result of the reduced Curie temperature with a negative voltage is not convincing because the measurement was not done at sufficiently high temperatures. We absolutely agree with the reviewer that the measurement at higher temperatures will show much more convincing results. However, it is not possible to measure at temperatures above 400 K due to the temperature constraint of the physical property measurement system (Quantum Design) we used. Nevertheless, even within the possible temperature range, there is a clear difference in the temperature dependence of R_H depending on the sign of V_G (Figure 2b of the main text). From the stronger temperature dependence in the sample with $V_G = -13$ V, we infer that the Curie temperature decreases with a negative voltage. The second question is whether the oxygen migration is temperature-dependent. The reviewer questions whether R_H is modified by the oxygen migration that may occur during the measurement at elevated temperature. We do not think so for two reasons; first, the gate voltage is not applied during the measurement. We first applied V_G to the top electrode and then measured R_H as a function of temperature with the gate floating. This is possible due to the non-volatile nature of the gate effect in our sample, which persists even after V_G is turned off. The second reason is that there is no significant change in magnetic moment under heating in the measurement temperature range. To confirm this argument, we measured the magnetic moment of the sample before and after heat treatment at 380 K. We used two samples of a W(4 nm)/CoFeB(2 nm)/AlO_x(1.5 nm) structure with plasma oxidation times of 0 s and 150 s. **Figures R8a and b** show the results that the magnetic moments of the samples do not change upon the heat treatment regardless of the plasma oxidation time. This result allows us to rule out the effect of oxygen migration on the temperature dependence of R_H .

We included the following sentences regarding magnetic moment change at 380 K on page 6 of the revised manuscript and Supplementary Note 3 as;

“Since the magnetic moment of the CoFeB does not change at the temperature up to 380 K (Supplementary Note 3), this result shows that the Curie temperature and J_{ex} of CoFeB decrease with a negative voltage.”

Figure R8. a, b Magnetic moment as a function of magnetic field in a W (4 nm)/CoFeB (2 nm)/AlO_x (2 nm) structure with different plasma oxidation time; **a.** 0 s and **b.**150 s, where the samples are heated for 5 minutes at 380 K.

8) The sentence "For the first process, lowering the damping of FM or reducing the thermal conductivity of FM enhances the SSE efficiency by several factors" (lines 50-52) is misleading. The thermal conductivity and "the first process" (i.e., SSE coefficient) are different parameters and can be designed independently.

Response) We apologize for the confusing sentence. Reflecting the reviewer’s comment, we have modified the sentence on page 3 of the revised manuscript as;

“Additionally, lowering the damping of FM for the first process or reducing the thermal conductivity of FM enhances the SSE efficiency by several factors.”

Reviewers' Comments:

Reviewer #1:

Remarks to the Author:

As stated in my first report, in my opinion the paper is sound, well written and should attract attention of the magnetism and spintronics community and deserves to be published in Nature Communications. I am happy with the authors reply to my question and the change made in the manuscript, thus I recommend its publication.

Reviewer #2:

Remarks to the Author:

As already stated in the first report:

The manuscript 'Enhanced spin Seebeck effect via oxygen manipulation' by Jeong-Mok Kim et al. reports a substantial enhancement of the Spin Seebeck Effect (SSE) by oxidizing a ferromagnet in normal metal/ferromagnet/oxide structures. In W/CoFeB/AlO_x structures the voltage induced interfacial oxidation of CoFeB modifies the SSE and results in the enhancement of thermoelectric signal by an order of magnitude which is the key finding of this manuscript and is a noteworthy result.

The experiments were extended based on feedback from referees, and the results support the original conclusions of the paper. My concerns regarding the theoretical modelling were addressed appropriately, and the increased heat conductivity due to oxidation was included in the data evaluation. Additionally, some previously missing literature has been incorporated.

As a result, the manuscript presents a proof of concept on how to enhance thermoelectric efficiency. The unjustified claims regarding absolute thermoelectric efficiency have been removed from the manuscript. While I believe that thermoelectric power generation has potential for niche applications, it is important to note that checking the temperatures of high and low temperature reservoirs often imposes strict limits on efficiency that are not typically discussed. However, since the manuscript does not discuss thermoelectric generators in general, I am satisfied with the removal or modification of these statements.

The paper presents a new mechanism for enhancing the spin Seebeck effect, which I believe is interesting to the scientific community. Therefore, I recommend that it be published in Nature Communications.

Reviewer #3:

Remarks to the Author:

The authors have carefully addressed all of my comments and the manuscript is improved. I thus recommend the publication of this paper in Nature Communications. However, before publication, the following issues should be addressed.

Following my previous comment 6), the authors have changed the definition of ΔV_{TE} and revised the longitudinal axes of Figs. 2e and 3a. However, the longitudinal axis of Fig. 3c in the revised manuscript is the same as that in the original version. Please recheck this point.

Figure S6b,c in Supplementary Information:

The longitudinal axis should be inverted; the upper (lower) half should be positive (negative).

Responses to the reviewers' comments

Reviewer #3:

The authors have carefully addressed all of my comments and the manuscript is improved. I thus recommend the publication of this paper in Nature Communications. However, before publication, the following issues should be addressed.

Following my previous comment 6), the authors have changed the definition of ΔV_{TE} and revised the longitudinal axes of Figs. 2e and 3a. However, the longitudinal axis of Fig. 3c in the revised manuscript is the same as that in the original version. Please recheck this point.

Response) First of all, we appreciate the reviewer's recommendation of our manuscript to publication in Nature Communications. To answer the question, we find that ΔV_{TE} in the Fig. 3c of the original version was already defined as the reviewer suggested, $\Delta V_{TE} = [V_{TE}(M//+x) - V_{TE}(M// -x)]/2$, therefore we do not modify the y axis of Fig. 3c in the revised manuscript. To confirm our statement, we compare Fig. 3c to Fig. 3b. In Fig. 3b, ΔV_{TE} is about 10 uV at $t = 0$ s (without oxidized ferromagnet lattices) and -10 uV at $t = 150$ s (with oxidized ferromagnet lattices), where the ΔV_{TE} is defined as the reviewer suggested. These ΔV_{TE} values are consistent to ΔV_{TE} of Fig. 3c, which is about 10 uV (without oxidized ferromagnet lattices at $t_{ox} = 0$ s and $V_G = +12$ V) and about -10 uV (with oxidized ferromagnet lattices at $t_{ox} = 150$ s and $V_G = -12$ V).

Figure S6b,c in Supplementary Information: The longitudinal axis should be inverted; the upper (lower) half should be positive (negative).

Response) We appreciate the reviewer's comment. We modified the axis of Figure S6 b and c in Supplementary Note 6 as suggested.